# The Knob Domain of the Fiber-1 Protein Affects the Replication of Fowl Adenovirus Serotype 4

**DOI:** 10.3390/microorganisms12112265

**Published:** 2024-11-08

**Authors:** Xiaofeng Li, Zhixun Xie, You Wei, Zhiqin Xie, Aiqiong Wu, Sisi Luo, Liji Xie, Meng Li, Yanfang Zhang

**Affiliations:** 1GuangXi Key Laboratory of Veterinary Biotechnology, GuangXi Veterinary Research Institute, Nanning 530000, China; lixiaofeng2003@126.com (X.L.); weiyou0909@163.com (Y.W.); xzqman2002@sina.com (Z.X.); wuaiqiong88@163.com (A.W.); 2004-luosisi@163.com (S.L.); xie3120371@163.com (L.X.); mengli4836@163.com (M.L.); zhangyanfang409@126.com (Y.Z.); 2Key Laboratory of China (GuangXi)-ASEAN Cross-Border Animal Disease Prevention and Control, Ministry of Agriculture and Rural Affairs of China, Nanning 530000, China

**Keywords:** FAdV-4, fiber-1, knob, innate immune signaling pathway, type I interferon, cytokines

## Abstract

Fowl adenovirus serotype 4 (FAdV-4) outbreaks have caused significant economic losses in the Chinese poultry industry since 2015. The relationships among viral structural proteins in infected hosts are relatively unknown. To explore the role of different parts of the fiber-1 protein in FAdV-4-infected hosts, we truncated fiber-1 into fiber-1-Δ1 (73–205 aa) and fiber-1-Δ2 (211–412 aa), constructed pEF1α-HA-fiber-1-Δ1 and pEF1α-HA-fiber-1-Δ2 and then transfected them into leghorn male hepatocyte (LMH) cells. After FAdV-4 infection, the roles of fiber-1-Δ1 and fiber-1-Δ2 in the replication of FAdV-4 were investigated, and transcriptome sequencing was performed. The results showed that the fiber-1-Δ1 and fiber-1-Δ2 proteins were the shaft and knob domains, respectively, of fiber-1, with molecular weights of 21.4 kDa and 29.6 kDa, respectively. The fiber-1-Δ1 and fiber-1-Δ2 proteins were mainly localized in the cytoplasm of LMH cells. Fiber-1-Δ2 has a greater ability to inhibit FAdV-4 replication than fiber-1-Δ1, and 933 differentially expressed genes (DEGs) were detected between the fiber-1-Δ1 and fiber-1-Δ2 groups. Functional analysis revealed these DEGs in a variety of biological functions and pathways, such as the phosphoinositide 3-kinase–protein kinase b (PI3K–Akt) signaling pathway, the mitogen-activated protein kinase (MAPK) signaling pathway, cytokine–cytokine receptor interactions, Toll-like receptors (TLRs), the Janus tyrosine kinase–signal transducer and activator of transcription (Jak–STAT) signaling pathway, the nucleotide-binding oligomerization domain (NOD)-like receptors (NLRs) signaling pathway, and other innate immune pathways. The mRNA expression levels of type I interferons (IFN-α and INF-β) and proinflammatory cytokines (IL-1β, IL-6 and IL-8) were significantly increased in cells overexpressing the fiber-1-Δ2 protein. These results demonstrate the role of the knob domain of the fiber-1 (fiber-1-Δ2) protein in FAdV-4 infection and provide a theoretical basis for analyzing the function of the fiber-1 protein of FAdV-4.

## 1. Introduction

Fowl adenovirus (FAdV) belongs to the adenovirus genus of the Adenoviridae family and can be divided into five genotypes (FAdV-A~FAdV-E) and 12 serotypes (1~7, 8a, 8b, 9~11) [1,2,3]. After the first FAdV infection occurred in Pakistan in 1987, subsequent outbreaks occurred worldwide, resulting in high mortality rates in chickens [4]. Once a flock is infected with FAdV, the virus spreads throughout the flock for a long time, posing a serious threat to the poultry industry worldwide [4,5]. In 2015, an outbreak of fowl adenovirus serotype 4 (FAdV-4) disease occurred in Shandong, China [6]. FAdV-4 is highly pathogenic in chickens, especially in 3~6-week-old broilers, and causes severe hydropericardial hepatitis syndrome (HHS). FAdV-4 primarily affects the heart and liver of chickens and can be isolated from the liver [6,7]. The typical symptoms include liver swelling, local necrosis and hemorrhage, cardiac cyst enlargement, and the presence of transparent yellow fluid [4]. Although FAdV-4 has been active for 30 years, few studies have examined the effects of its viral structural proteins on infected hosts.

FAdVs are nonenveloped double-stranded DNA viruses with virion diameters of 70~90 nm and icosahedral structures, and each virion is composed of 252 capsids [8]. Each capsid contains the major structural proteins hexon, penton base, and fiber and the minor proteins X, VI, VII, and VIII. The hexon protein is often targeted in molecular epidemiology studies because of its epitope and serum neutralizing properties [9]. During infection, the penton base binds to host cells and relies on host cell surface integrins to mediate endocytosis [10], and studies have shown that the use of the penton base as a subunit vaccine in 2-week-old SPF chickens provides a 90% protection rate against infection, indicating that the penton base is a candidate protein for subunit vaccines [10].

The fiber protein consists of three domains, the knob, shaft and tail, and varies among serotypes. FAdV-1, FAdV-4, and FAdV-10 have two fibrous proteins, fiber-1 and fiber-2, which play different roles in the infection process of FAdV-4. Recent studies have shown that fiber-1, but not fiber-2, directly causes FAdV-4 infection [11,12]. However, in highly pathogenic FAdV-4 strains, fiber-2 and hexon, rather than fiber-1, are virulence determinants [13,14,15]. Studies have shown that during FAdV-4 infection, the fiber-1 protein attaches the viral capsid to the host cell surface through its knob domain interaction with the D2 domain of the CAR cell receptor to mediate viral infection [11]. Fiber-2 promotes viral replication. The tail of the fiber-1 protein is linked to the penton base. The shaft and knob domains of fiber-1 are considered key factors in viral infection [16]. The use of serum antibodies against ffiber-1 is effective enough to neutralize the virus and prevent FAdV-4 infection [12].

After the virus infects the host, the innate immune system is activated to recognize the virus through numerous pattern recognition receptors (PRRs). The PRRs include Toll-like receptors, melanoma differentiation-associated gene 5 (MDA5), and cyclic guanosine monophosphate-adenosine monophosphate synthase (cGAS) receptors [17,18]. PRRs recognize viral nucleic acids and activate downstream signaling pathways to induce the production of type I interferons (IFN-α/β) and inflammatory cytokines (IL-1β, IL-6, IL-8, and IL-15) to inhibit viral replication. However, FAdV-4 can evade the innate immune response of the host and even develop various defense mechanisms to ensure its own survival and replication. Proteins often perform their physiological functions by forming protein complexes or interacting with nucleic acids. During viral replication, the interaction between viral proteins and host proteins plays a key role in disease development and prognosis.

The results of our previous study revealed that fiber-1 protein overexpression in LMH cells after infection with FAdV-4 significantly upregulated the mRNA expression of TLR receptors (TLR1b, TLR3, TLR7, and TLR21) and related signaling pathways (myeloid differentiation factor 88 (MyD88), IRF7, and IFN-β) [19]. In this study, we cloned fiber-1-Δ1 (73–205 aa) and fiber-1-Δ2 (211–412 aa) to construct eukaryotic expression vectors, investigated their effects on FAdV-4 replication, and analyzed their molecular mechanisms via transcriptome sequencing and real-time quantitative fluorescence PCR (qPCR). The results of this study provide a basis for understanding the structure and function of the fiber-1 protein and a reference for further elucidating the pathogenic mechanism of FAdV-4 and host immune stress during infection.

## 2. Materials and Methods

### 2.1. Virus and Cells

The FAdV-4 virus strain used in this study was isolated from the liver of a chicken infected with HHS in Nanning, Guangxi, and maintained at the Guangxi Veterinary Research Institute. LMH cells were cultured in Dulbecco’s modified Eagle’s medium (DMEM)/F-12 (Gibco, Grand Island, NY, USA, Code No. C11330500BT) supplemented with 10% fetal bovine serum (FBS; Gibco, Code No. 10099141C1) and 1×penicillin–streptomycin mixture (Solarbio, Beijing, China; Code No. P1400).

### 2.2. Identification of Fiber-1 Protein Domains and Their Functionality via Prediction Software

The domain of the fiber-1 protein was predicted via the online software SMART (version 9) (http://smart.embl-heidelberg.de/, accessed on 24 August 2023). The prediction results are given below. According to the functional annotation of the truncated protein, 73–205 aa (fiber-1-Δ1) and 211–412 aa (fiber-1-Δ2) were selected for the experiment. The antigenic peptide fragments of fiber-1-Δ1 and fiber-1-Δ2 were predicted by DNAMan version 6, and the hydrophilicity, antigenicity, flexibility and surface accessibility of the potentially dominant antigenic epitopes in B cells were predicted by DNAStar Protean version 7.1.0.

### 2.3. Cloning Genes and Construction Plasmids

The primers were synthesized with reference to two gene sequences. The fiber-1-Δ1 and fiber-1-Δ2 genes were cloned from DNA extracted from the FAdV-4 virus, separately inserted into the pEF1α-HA vector, and verified by sequencing by RayBiotech (Guangzhou, China). The specific primers used for cloning the fiber-1-Δ1 and fiber-1-Δ2 genes are listed in Table 1.

### 2.4. Transfection and Verification by Western Blotting

LMH cells were cultured in 6-well plates, followed by transfection with pEF1α-HA (mock), pEF1α-HA-fiber-1-Δ1 and pEF1α-HA-fiber-1-Δ2 plasmids (1, 1.25, or 2.0 µg per well) via a Lipofectamine^TM^ 3000 Transfection Kit (Invitrogen, Carlsbad, CA, USA; Code No. L3000015). Twenty-four hours after transfection, in our previous experiments, the TCID50 of this strain was known to be 10^−7.4^/mL [19]. Cells were infected with FAdV-4 (MOI = 0.01), and cell samples and culture medium supernatants were collected and lysed with lysis buffer 24 h later. The cell samples were mixed with 1× sodium dodecyl sulfate (SDS) loading buffer (Solarbio, Code No. P1043), boiled and then incubated on ice. Protein samples were separated by sodium dodecyl sulfate polyacrylamide gel electropheresis (SDS–PAGE, Solarbio, Code No. PG01010) and transferred to polyvinylidene fluoride (PVDF) membranes. The membranes were incubated with Western blot blocking solution (Solarbio, Code No. P0023B) at 4 °C overnight due to a lack of antibodies against the fiber-1-Δ1 and fiber-1-Δ2 proteins and then incubated with an anti-HA monoclonal antibody (Invitrogen, Code No. 26183). The primary antibody was discarded, and the membrane was washed three times (10 min each time) with 1× phosphate-buffered saline with Tween-20 (PBST, Solarbio, Code No. 1033), incubated with goat anti-mouse IgG antibody (Invitrogen, Code No. F-2761) for 1 h, and washed four times with 1× PBST buffer. Before imaging, the membranes were incubated with a 3,3′-diaminobenzidine (DAB) horseradish peroxidase color development kit (Beyotime Biotechnology, Beijing, China, Code No. P0202) working solution for 3–10 min at room temperature.

### 2.5. Immunofluorescence Assay

After the pEF1α-HA-fiber-1(plasmids kept in laboratory), pEF1α-HA-fiber-1-Δ1 and pEF1α-HA-fiber-1-Δ2 plasmid transfection and FAdV-4 infection, the culture medium supernatant was removed, and the cells were incubated sequentially with 4% paraformaldehyde (Solarbio, Code No. P1110) for 30 min at room temperature, Triton X-100 (Solarbio, Code No. T8200) for 30 min, and 5% bovine serum albumin (BSA) blocking buffer (Solarbio, Code No. SW3015) for 1 h. The cells were then incubated with an anti-HA monoclonal antibody (Invitrogen, Code No. Code No. 26183) at 4 °C overnight. After removal of the primary antibody, the cells were washed with PBS, incubated with a fluoresceine isothiocyanate (FITC)-conjugated rabbit anti-mouse antibody (Invitrogen, Code No. F-2761), and stained with 4′,6-diamidino-2-phenylindol (DAPI, Solarbio, Code No. C0065). The DAPI was discarded, the cells were washed three times with PBS, and the results were observed via inverted fluorescence microscopy and laser confocal microscopy.

### 2.6. Measurement of FAdV-4 Growth in LMH Cells

After the plasmid transfection and FAdV-4 infection, the cells were harvested at different time points (6, 12, 18, 24, 36, and 48 hpi). Viral DNA was extracted via a Universal Genomic DNA Kit (Cowin Biotechnology, Beijing, China, Code No. CW2298M). The DNA concentration was normalized to 100 ng/µL. qPCR was performed with 2×SYBR Green Mix (10 µL, Thermo Fisher Scientific, Boston, MA, USA, Code No. A257420), 10 μmol/L hexon-U and hexon-D primers (1 µL, Table 2), a DNA template (2 µL), and RNase-free water (6 µL, Solarbio, Code No. R1600). The absolute quantitative PCR program was as follows: 94 °C for 2 min and 40 cycles of 94 °C for 15 s and 60 °C for 30 s. This procedure was repeated 3 times for each sample. The results were calculated via a standard curve (y = −3.362x + 36.081; R^2^ = 0.999) previously established by our laboratory for the detection of the FAdV-4 viral load. The data are expressed as the mean ± SD.

### 2.7. Transcriptome Sequencing Analysis

The cells were harvested 48 h post-transfection and sent to Sangon Biotech (Shanghai, China) for transcriptome sequencing. DESeq2 (version 1.12.4) was used to determine the DEGs between two samples. The gene expression differences were visualized via volcano plots.

### 2.8. Functional Analysis of DEGs

To functionally classify the DEGs, the DEGs were subjected to Gene Ontology (GO) and Kyoto Encyclopedia of Genes and Genomes (KEGG) analyses. The GO and KEGG analyses were performed with the assistance of Shanghai Sangon Biotech.

### 2.9. RNA Extraction and qPCR

After transfection, the cells were infected with FAdV4 at an MOI of 0.01, and the cell cultures were collected at 48 h. The RNA was extracted via the Gene JET RNA Purification Kit (Invitrogen, Code No. K0732). After the concentration was determined, the RNA was reverse transcribed into cDNA, and the reaction mixture was 20 μL: 4 μL of 5×PrimeScript RT Master Mix (Takara Bio Inc., Beijing, China; Code No. RR036A), 1 μg of RNA, and nuclease-free water. The reverse transcription procedure was performed as follows: 37 °C for 15 min and 85 °C for 5 s. The resulting cDNA was diluted 10-fold and used as a template for the qPCR. The qPCR setup and reaction procedure are described above in Section 2.6. Three replicates were performed for each sample. GAPDH was used as the reference gene. The expression levels of TLR3, TRIF, IRF7, IFN-α, INF-β, IL-1β, IL-6 and IL-8 were measured, and the primer sequences are shown in Table 2. The expression levels of the genes were calculated via the 2^−ΔΔCt^ method.

### 2.10. Data Processing and Statistical Analysis

The data were plotted with GraphPad Prism 8.0 software. Differences between groups were analyzed by the *t* test with IBM SPSS Statistics version 29. Here, * indicates *p* < 0.05 and ** indicates *p* < 0.01.

## 3. Results

### 3.1. Identification of Fiber-1 Protein Domains and Cloning of the Genes

The predicted results are shown below (Figure 1A). According to the functional annotation of the truncated protein, 73–205 aa (fiber-1-Δ1) and 211–412 aa (fiber-1-Δ2) were selected for the experiment (Figure 1B). According to the three parts of fiber-1, knob, shaft and tail, fiber-1-Δ1 was the shaft domain of fiber-1 according to GenBank accession no. MH392486.1, and fiber-1-Δ2 was the knob domain of fiber-1 according to GenBank accession no. MH392487.1. The fiber-1-Δ1 and fiber-1-Δ2 gene segments were amplified via DNA extracted from viruses as a PCR template. The PCR amplification products were separated on an agarose gel, with bands at approximately 399 bp and 606 bp (Figure 1C), which were consistent with the expected results.

We used DNAMan to predict the antigenic peptide fragments of fiber-1-Δ1 and fiber-1-Δ2, and as shown in Table 3, there were 5 antigenic peptide fragments in fiber-1-Δ1 and 10 antigenic peptide fragments in fiber-1-Δ2. The hydrophilicity, antigenicity, flexibility and surface accessibility of the potentially dominant antigenic epitopes in the B-cells are shown in Figure 1D.

### 3.2. Expression of Fiber-1-Δ1 and Fiber-1-Δ2 in LMH Cells

In the FAdV-4-infected cells at an MOI of 0.01, the virus proliferated rapidly at 12~48 h, then proliferated slowly, and reached peak growth at 96 h (Figure 2A). To investigate the optimal transfection efficiency, we used different transfection plasmid concentrations (1, 1.25, and 2.0 µg per well). The results revealed that the transfection efficiency increased with an increasing concentration of the transfected plasmid (Figure 2B). On the basis of the test results and the plasmid concentration recommended by the kit, the cells were transfected with 2.0 µg/well.

After the recombinant plasmids pEF1α-HA-fiber-1-Δ1 and pEF1α-HA-fiber-1-Δ2 were transfected into LMH cells, the fiber-1-Δ1 and fiber-1-Δ2 proteins were verified by Western blotting. Owing to the lack of fiber-1-Δ1 and fiber-1-Δ2 monoclonal antibodies, anti-HA monoclonal antibodies were used to verify their expression in LMH cells. The results revealed specific bands of approximately 21.4 kDa and 29.6 kDa (Figure 2C) in the cells transfected with pEF1α-HA-fiber-1-Δ1 and pEF1α-HA-fiber-1-Δ2, whereas specific bands were not detected in the cells transfected with pEF1α-HA (mock). To further verify that the fiber-1-Δ1 and fiber-1-Δ2 proteins were correctly expressed in LMH cells, the subcellular localization of the fiber-1-Δ1 and fiber-1-Δ2 proteins in the cells was analyzed by immunofluorescence and laser confocal microscopy. The cell nuclei were labeled with blue fluorescence, and the recombinant proteins were labeled with green fluorescence. As shown in Figure 2D, green fluorescence was mainly observed in the cytoplasm of LMH cells, whereas control cells transfected with the pEF1α-HA vector did not show green fluorescence, indicating that the fiber-1-Δ1 and fiber-1-Δ2 proteins were localized in the cytoplasm of LMH cells.

To confirm that fiber-1-Δ1 and fiber-1-2 proteins were localized in the cytoplasm of LMH cells, we simultaneously transfected the full-length fiber-1 protein into LMH cells, and the results showed that the fiber-1 protein was expressed in the cytoplasm (Figure 2D). Meanwhile, the amino acid sequences of fiber-1, fiber-1-Δ1 and fiber-1-Δ2 were analyzed online (https://www.genscript.com/psort.html, accessed 24 August 2024); the results showed that there were no nuclear localization signals in the fiber-1-Δ1 and fiber-1-Δ2 protein sequences. fiber-1, fiber-1-∆1 and fiber-1-∆2 were predicted to be expressed in the cytoplasm by Reinhardt’s cytoplasmic/nuclear discrimination method with a reliability of 70.6%, 94.1% and 76.7%, respectively (Figure 2E).

### 3.3. High-Level Expression of Fiber-1-Δ1 and Fiber-1-Δ2 Inhibits FAdV-4 Replication

To investigate the effects of fiber-1-Δ1 and fiber-1-Δ2 on FAdV-4 replication in LMH cells, LMH cells were transfected with pEF1α-HA-fiber-1-Δ1 and pEF1α-HA-fiber-1-Δ2 plasmids and then infected with FAdV-4. At different time points (6, 12, 18, 24, 36, and 48 h) after FAdV-4 infection, viral replication was determined via absolute quantitative PCR. The results are calculated by log10/μL, as shown in Figure 3 below.

When the cells were transfected with pEF1α-HA, the virus grew rapidly. Compared with those in the mock group, the number of LMH cells infected with pEF1α-HA-fiber-1-Δ1 was lower, but the effect was not significant (*p* > 0.05). The LMH cells transfected with pEF1α-HA-fiber-1-Δ2 had substantially reduced viral loads from 24 hpi (*p* < 0.05) to 36 hpi (*p* < 0.05) to 48 hpi (*p* < 0.01), suggesting that compared with the fiber-1-Δ1 protein, the fiber-1-Δ2 protein has a strong ability to inhibit FAdV-4 replication (Figure 3).

### 3.4. Identification and Clustering of DEGs

The aforementioned results showed that fiber-1-Δ2 has a more obvious inhibitory effect on viruses than fiber-1-Δ1. Next, the underlying molecular mechanisms and functional differences between these two proteins were explored.

After LMH cells were transfected with pEF1α-HA-fiber-1-Δ1, pEF1α-HA-fiber-1-Δ2, or FAdV-4 for 48 h, a total of 933 DEGs were identified between the pEF1α-HA-fiber-1-Δ1 group (A) and the pEF1α-HA-fiber-1-Δ2 group (B). In the pEF1α-HA-fiber-1-Δ1 group (A), compared with the pEF1α-HA-fiber-1-Δ2 (B) group, 194 of these genes were upregulated and 739 genes were downregulated (Figure 4).

### 3.5. GO Enrichment and KEGG Enrichment Analyses

To further analyze the DEGs, we classified the genes into functional categories. GO analysis of the DEGs was conducted to elucidate the functions of the DEGs. The DEGs were annotated on the basis of the GO biological process, cellular component, and molecular function categories. The 30 most enriched GO terms (*p* < 0.05) are shown in Figure 5A. The GO analysis revealed that the DEGs were enriched in cell communication, response to chemicals, cellular response to stimulus, negative regulation of biological processes, regulation of multicellular organisms, and response to organic substances.

The DEGs were subjected to Kyoto Encyclopedia of Genes and Genomes (KEGG) pathway enrichment analysis to explore the biological processes and molecular functions involved in FAdV-4 infection. As shown in Figure 5B, 30 KEGG pathways (*p* < 0.05) were enriched among the DEGs between the two groups. Among the 30 KEGG pathways, 6 were associated with immune function, such as the tumor necrosis factor (TNF) signaling pathway, the transforming growth factor-beta (TGF-beta) signaling pathway, the PI3K–Akt signaling pathway, the MAPK signaling pathway, cytokine–cytokine receptor interactions, and the B-cell receptor signaling pathway.

### 3.6. Activation of Several Immune-Related Pathways After FAdV-4 Infection

The genes enriched in canonical pathways after fiber-1-Δ1 and fiber-1-Δ2 overexpression in LMH cells are listed in Table 4. These canonical pathways are mainly related to the host immune response. The RIG-I-like helicase receptors (RIG-I-like receptor) signaling pathway, NOD-like receptor signaling pathway, Toll-like receptor (TLR) signaling pathway, Jak–STAT signaling pathway, and cytokine–cytokine receptor interaction pathway are associated with the innate immune response and inflammatory response. Compared with those in the fiber-1-Δ1 overexpression group, the results revealed that, in the RIG-I-like receptor signaling pathway, only IL8L2 was downregulated, whereas in the NOD-like receptor signaling pathway, receptor-interacting protein kinase 3 (RIPK3) and IL8L2 were downregulated. FOS, JUN, PIK3R3, IL8L2 and CCL4 were downregulated in the Toll-like receptor signaling pathway. CISH, CSF3R, PIK3R3, OSMR, CDKN1A, SOCS3, SOCS2, PIM1, and STAT3 were downregulated in the Jak–STAT signaling pathway. BMP7, TNFRSF21, CSF3R, PDGFB, OSMR, IL8L2, TNFRSF8, TGFB1, TGFB2, TNFSF15, CCL4, CXCL14 and LOC10705392 were downregulated in the cytokine–cytokine interaction pathway. Some genes were enriched in one or more pathways. Only the PI3K–Akt signaling pathway, MAPK signaling pathway, and cAMP signaling pathway had upregulated genes, and upregulated genes were not detected in the other pathways. These results suggest that fiber-1-Δ2 inhibits FAdV-4 replication possibly by activating innate immunity during FAdV-4 infection.

### 3.7. Effects on Type I Interferons and Cytokines

According to the above results, the fiber-1-Δ1 and fiber-1-Δ2 proteins exert different inhibitory effects on the replication of FAdV-4 through different innate immune signaling pathways. To further explore and validate the role of fiber-1-Δ1 and fiber-1-Δ2 in the FAdV-4 infection process, the relative mRNA transcript levels of TLR3, TLR7, TRIF, IRF7, IFN-α, IFN-β, IL-1β, IL-6 and IL-8 were measured via qPCR.

The levels of TLR3, TLR7, TRIF, IFN-α, IFN-β, IL-1β and IL-6 were greater in fiber-1-Δ1-overexpressing cells than in control cells, but the difference was not significant (*p* > 0.05). Compared with that in the control group, the level of IRF7 in the fiber-1-Δ1-overexpressing group was lower, but the difference was not significant (*p* > 0.05).

Compared with those in the control group, in the fiber-1-Δ2 group, the levels of TLR3, IFN-β and IL-8 were significantly increased (*p* < 0.05), and the levels of TIRF, IFN-α and IL-1β were significantly increased (*p* < 0.01). The levels of IRF7 and IL-6 were significantly increased, but the differences were not significant (*p* > 0.05).

Compared with those in fiber-1-Δ1-overexpressing cells, IFN-β, IL-6, IL-8 (all *p* < 0.05), IFN-α and IL-1β (both *p* < 0.01) were significantly upregulated in fiber-1-Δ2-overexpressing cells. The levels of other mRNAs were not significantly different between the groups (*p* > 0.05) (Figure 6).

## 4. Discussion

FAdV-4 infection can lead to cases of severe HHS in chickens. The high number of cases of FAdV-4 infection on farms has caused considerable economic losses and hindered the development of the poultry industry [6,20]. To develop highly effective novel HHS vaccines, a complete understanding of FAdV-4 infection is needed. However, the mechanisms of FAdV infection have not yet been determined.

Fiber-1, an adenoviral spike protein of FAdV-4, plays a crucial role in mediating viral infection and inducing neutralizing antibodies [21,22]. Previous studies have shown that the fiber-2 protein acts as a subunit vaccine, and a single immunization of 10-day-old chickens induces humoral and cellular immunity that is protective against the virus but does not induce neutralizing antibody production [23]. The fiber proteins of FAdV-4 and FAdV-11 were transformed into a chimeric subunit vaccine, which was subsequently used to immunize SPF chickens, which were then challenged with strains that cause HHS or inclusion body hepatitis (IBH), and the development of neutralizing antibodies limited against FAdV-11 and absent against FAdV-4 indicated that the protection conferred by such an antigen may be linked to different immunization pathways [24]. Moreover, studies have reported that when fiber-1 was used as a subunit vaccine, the first immunization of 2-week-old chickens did not induce neutralizing antibodies; rather, neutralizing antibodies appeared after the second immunization of 4-week-old chickens, and antibody production was maintained for 10 weeks. Therefore, the age of the chickens immunized with FAdV-4 fibrin may affect the production of neutralizing antibodies [25]. The fiber-1 protein is divided into tail, shaft, and knob segments. In this study, fiber-1-Δ1 was the shaft domain of fiber-1, and fiber-1-Δ2 was the knob domain of fiber-1. The shaft domain of fiber-1 represents the N-terminal domain of avian adenovirus fiber proteins, which have been linked to variations in virulence [26]. Avian adenoviruses possess penton base capsomers that consist of a pentameric base associated with two fibers [27]. The shaft of the fiber plays an important role in the entry of adenovirus into host cells, primarily by facilitating the binding of fibers to cell receptors, as well as the interaction of the penton base with cellular integrins [28]. The knob domain of fiber-1 represents the C-terminal part of the head domain of the dsDNA viruses, not the RNA-stage adenovirus. This is a globular head domain with an antiparallel beta-sandwich fold formed by two four-stranded beta-sheets with the same overall topology as human adenovirus fiber heads. This C-terminal domain is the receptor-binding domain of the avian adenovirus long fiber [29]. Moreover, many adenoviruses (including human adenoviruses and avian adenoviruses) contain the spike protein. The knob domain contains many antigenic sites and has an epitope against specific neutralizing antibodies. Studies have shown that serum targeting the knob domain can block FAdV-4 infection and that a fusion protein containing the knob domain has an effective protective effect on a lethal attack of FAdV-4 in chickens [12]. In our study, we utilized in silico methods to predict the epitopes/peptides of fiber-1-Δ1 and fiber-1-Δ2. These antigenic sites, or epitopes, are specific chemical groups on the surface of antigen molecules that activate the immune system and facilitate antibody interactions. We evaluated key characteristics of the potentially dominant B-cell epitopes for both fiber-1-Δ1 and fiber-1-Δ2, including hydrophilicity, antigenicity, flexibility, and surface accessibility. Our findings may provide ideas to explain the inhibition of viral replication by fiber-1-Δ2 in terms of natural immunity, as well as valuable insights for the development of novel diagnostic methods and related vaccines. However, it is important to note that this in silico approach is less definitive than in vitro or in vivo testing, which represents a limitation of this study.

To test the role of two fragments of the fiber-1 protein in FAdv-4 replication, we transfected LMH cells with pEF1α-HA-fiber-1-Δ1 and pEF1α-HA-fiber-1-Δ2 and then infected them with FAdV-4. The LMH cell line is an epithelial cell line of chicken hepatocellular carcinoma [30]. It once served as an analytical model for transcriptomic studies on hosts infected with FAdV-4 [31,32]. These findings suggest that LMH cells can serve as a suitable model for investigating FAdV-4 infection and host–FAdV-4 interactions.

In addition, the dynamics of virus replication at 6, 12, 18, 24, 36, and 48 hpi after FAdV-4 infection were determined via an absolute relative quantification technique, and the results revealed that fiber-1-Δ1 and fiber-1-Δ2 inhibited the replication of FAdV-4 in LMH cells. Fiber-1-Δ2 had a more obvious inhibitory effect on viruses than fiber-1-Δ1, and the viral load of the cells in each group did not differ at 6~18 h. We hypothesized that it took a period for the virus to enter the host cells and that the host had not yet responded. After 24 h, the viral load of the cells in the fiber-1-Δ2 group was significantly lower than that in the control group, possibly because fiber-1-Δ2 stimulated the innate immune immunity of LMH cells. High-throughput RNA-seq analysis was performed on leghorn male hepatocytes (LMHs) at 12, 24, and 48 h after LMF infection with FAdV-4, and the top 10 GO categories were significantly enriched in biological processes without a response to stimulus and in the cellular response to stimulus not detected in the analysis [31]. The roles of fiber-1-Δ2 and fiber-1-Δ1 in the replication of FAdV-4 differ and are worthy of further exploration.

To explore the reasons for the above results, LMH cells were transfected with pEF1α-HA-fiber-1-Δ1 or pEF1α-HA-fiber-1-Δ2, and the cells were collected after FAdV-4 infection for transcriptome sequencing. A total of 933 DEGs were identified; compared with those in fiber-1-Δ2-overexpressing cells, 194 of these genes were upregulated and 739 of these genes were downregulated in fiber-1-Δ1-overexpressing cells. Further analysis revealed that these DEGs were involved in a range of biological processes, including cell communication, response to chemicals, negative regulation of biological processes, cellular response to stimuli, regulation of multiple organelles, and response to organic substances. The majority of the genes were involved in cell communication, response to chemicals, and the cellular response to stimuli. These results indicate that fiber-1-Δ1 and fiber-1-Δ2 have different responses to FAdV-4 infection in LMH cells. Further analysis revealed that the following signaling pathways were enriched in the DEGs: focal adhesion, the TNF signaling pathway, the TGF-beta signaling pathway, the PI3K–Akt signaling pathway, the MAPK signaling pathway, cytokine–cytokine receptor interaction, the B-cell receptor signaling pathway, and axon guidance.

The activated immune signaling pathways were further analyzed. The RIG-I-like receptor signaling pathway, NOD-like receptor signaling pathway, TLR signaling pathway, Jak–STAT signaling pathway, and cytokine–cytokine receptor interaction pathway were associated with the innate immune response and inflammatory response.

Innate immunity is a general protective mechanism that has evolved in organisms as the first line of defense against pathogens, and innate immunity is the basis of acquired immunity. Pathogen-associated molecular patterns (PAMPs), such as the nucleic acids of viruses, are first recognized by the PRRs of host cells and induce a series of signaling cascades that limit the proliferation and spread of the virus. During the process of virus infection in host cells, nucleic acids can be recognized by RIG-I-like receptors, NOD-like receptors and TLRs. The transcriptome sequencing results revealed differences in gene expression across different signaling pathways: in the fiber-1-Δ1 group compared with the fiber-1-Δ2 group, in the RIG-I-like receptor signaling pathway, only IL8L2 was downregulated, and in the NOD-like receptor signaling pathway, RIPK3 and IL8L2 were downregulated.

Compared with those in the fiber-1-Δ2 group, in the fiber-1-Δ1 group, the TLR signaling pathway, FOS, JUN, PIK3R3, IL8L2 and CCL4 were downregulated. In the Jak–STAT signaling pathway, CISH, CSF3R, PIK3R3, OSMR, CDKN1A, SOCS3, SOCS2, PIM1 and STAT3 were downregulated. With respect to the cytokine–cytokine receptor interactions, BMP7, TNFRSF21 CSF3R, PDGFB, OSMR, IL8L2, TNFRSF8, TGFB1, TGFB2, TNFSF15, CCL4, CXCL14, and LOC10705392 were downregulated. Some genes were enriched in one or more pathways, suggesting that these genes play a role in multiple pathways. These results suggest that FAdV-4 infection activates innate immune and inflammatory responses and that the virus uses the material of the host for replication while also being inhibited by the host.

TLRs play crucial roles in the innate immune response by recognizing highly conserved microbial structural motifs, including PAMPs, and by activating a series of downstream signals that induce the secretion of inflammatory cytokines and type I interferons [33]. Ten TLRs have been identified in chickens [34,35,36,37]. TLR3 is an intracellular TLR that is expressed at varying levels in the liver, kidney, cecal tonsils, and intestines, as well as in CD8+ and TCR1 cells [38]. Mammalian TLR3 can be activated by viral dsRNA or poly(I:C) analogs, which bind to each other to activate the adaptor protein TRIF [33,39] and subsequently promote the production of interferon through signal transduction, promoting viral infection and replication. In our study, compared with those of the control cells, the mRNA transcript levels of TLR3 and TIRF were significantly increased in fiber-1-Δ2-overexpressing cells. The level of TLR3 significantly increased in these cells, which is consistent with the finding of Zhang et al. [22] that FAdV-4 infects LMH cells and causes TLR3 upregulation. Like TLR3, TLR7 is an intracellular TLR that is expressed in a variety of tissues and cells. Stimulation with TLR7 receptor agonists (R848, poly(U)) upregulates the expression of the inflammatory cytokines IL-1β and IL-8 and type I interferon [35,40]. In our study, the TLR7 levels were increased in the fiber-1-Δ1 and fiber-1-Δ2 groups, but the difference was not significant. It is hypothesized that fiber-1-Δ2 activates downstream type I interferon and cytokine expression through the TLR3 and TIRF signaling pathways.

IFNs and other innate immune-related factors constitute the first line of host defense against pathogenic infection. They activate factors that aid in inhibiting viral replication and transmission. Compared with those in the fiber-1-Δ1 group, the levels of IFN-β, IL-6 and IL-8 were significantly upregulated (*p* < 0.05) in the fiber-1-Δ2 group. Similarly, the levels of IL-6 and IL-8 were consistent with those reported in a study of FAdV-4 infection in SPF chickens by Li Rong [41], indicating that our results are reliable. These results also suggest that fiber-1-Δ2 inhibits FAdV-4 replication by upregulating these cytokines. Compared with those in the fiber-1-Δ2 group, the levels of IFN-α and IL-1β were significantly greater in the fiber-1-Δ2 group (*p* < 0.01). Infection with the highly pathogenic FAdV-4 has been shown to induce inflammatory damage in many tissues, accompanied by high secretion levels of the proinflammatory cytokine IL-1β [41,42,43]. In this study, overexpression of the fiber-1-Δ2 protein upregulated the expression of type I interferon (IFN-α, IFN-β), IL-1β, IL-6 and IL-8 during FADV-4 infection of LMH cells and inhibited viral replication.

From the above results, it can be seen that fiber-1-Δ2 activated the natural immune signaling pathway to inhibit the replication of FAdV-4 in LMH cells. In this experiment, fiber-1-Δ2 was obtained by truncating fiber-1, which is rich in B-cell antigenic epitopes, the key to successful vaccine development. At the same time, the truncated protein can effectively avoid unnecessary antigenic components and reduce potential toxic reactions. Meanwhile, fiber-1-Δ2 may have better antigenicity than full-length fiber-1 and can be used as a detection antigen for the development of the FAdV-4 ELISA antibody assay with higher sensitivity to react with specific antibodies, the development of more valuable assays and more to understand all the aspects of viral pathogenesis in order to propose novel antiviral strategies. Finally, fiber-1-Δ2 is much smaller than full-length fiber-1, which facilitates the in vitro expression of FAdV-4 used to study the FAdV-4–host relationship.

In conclusion, our results showed that fiber-1-Δ2 (211–412 aa), which is the knob domain of fiber-1, significantly inhibits virus replication by activating innate immune signaling pathways. These findings lay a theoretical foundation for the analysis of the function of the fiber-1 protein of FAdV-4.

## Figures and Tables

**Figure 1 microorganisms-12-02265-f001:**
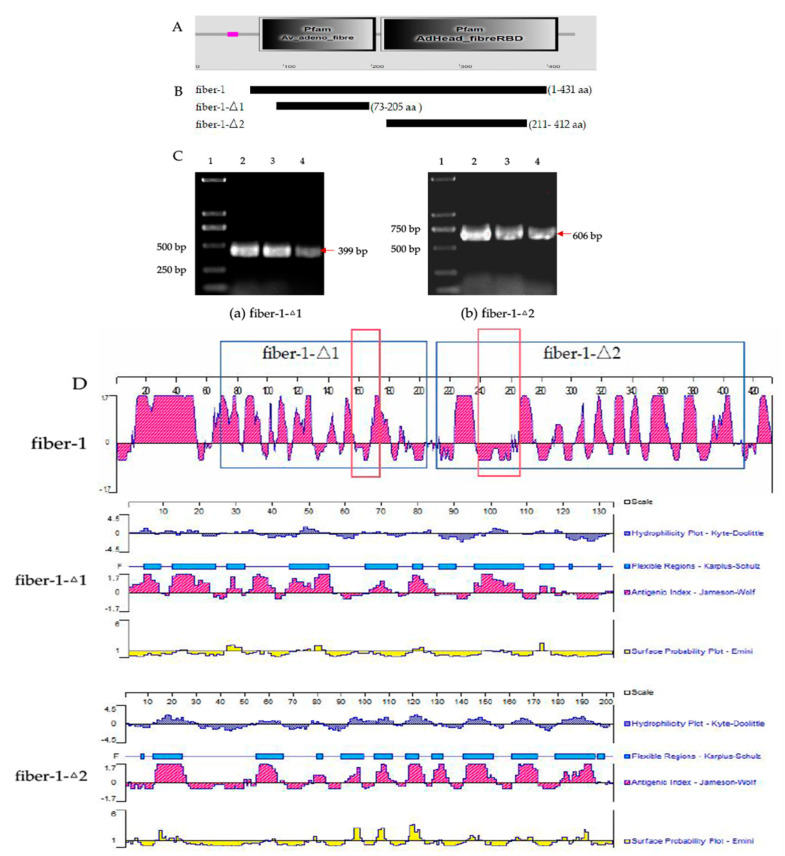
Prediction of fiber-1 major structural domains, PCR amplification and antigenic peptide prediction analysis. (**A**) Prediction of the fiber-1 protein domain. (**B**) Schematic of the amino acids of the full-length or truncated fiber-1 molecule, fiber-1 (1–431 aa), fiber-1-Δ1 (73–205 aa), and fiber-1-Δ2 (211–412 aa). (**C**) Analysis of the PCR products of the fiber-1-Δ1 and fiber-1-Δ2 genes via agarose gel electrophoresis. (**a**) The results for fiber-1-Δ1. Lane 1: DL2 000 DNA marker. Lanes 2~4: fiber-1-Δ1 gene PCR amplification products. The size of the fiber-1-Δ1 gene fragment is 399 bp. (**b**) The results for fiber-1-Δ2. Lane 1: DL2 000 DNA marker; lanes 2~4: fiber-1-Δ2 gene PCR amplification products. The size of the fiber-1-Δ2 gene fragment was 606 bp. (**D**) Complete fibre-1 protein map and prediction of the hydrophilicity, antigenicity, flexibility and surface accessibility of potential dominant B-cell epitopes of fiber-1-Δ1 and fiber-1-Δ2. Red boxes represent the highest scoring antigenic epitopes.

**Figure 2 microorganisms-12-02265-f002:**
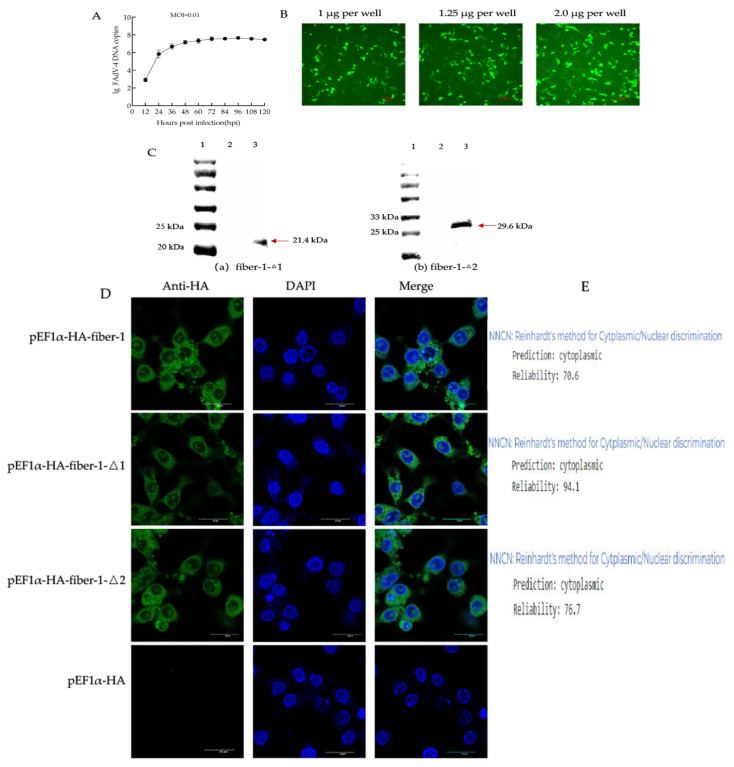
Fiber-1-Δ1 and fiber-1-Δ2 were expressed in LMH cells. (**A**) Replication curves of FAdV-4 on LMH, MOI = 0.01. (**B**) LMH cells were transfected with different plasmid concentrations (10×). The plasmid concentrations were 1, 1.25, and 2.0 µg per well, and the transfection efficiency increased with increasing concentration of the transfected plasmid. (**C**) The results for fiber-1-Δ1 and fiber-1-Δ2 were verified by Western blotting. (**a**) The results of fiber-1-Δ1. Lane 1: protein marker (11–245 kDa). Lane 2: LMH cells transfected with the empty pEF1α-HA plasmid as a control. Lane 3: fiber-1-Δ1 protein was overexpressed in LMH cells. The size of the fiber-1-Δ1 protein containing the HA label was approximately 21.4 kDa. (**b**) The results for fiber-1-Δ2. Lane 1: protein marker (11–245 kDa). Lane 2: LMH cells transfected with the empty pEF1α-HA plasmid as a control. Lane 3: fiber-1-Δ2 protein was overexpressed in LMH cells. The size of the fiber-1-Δ2 protein containing the HA label was approximately 29.6 kDa. (**D**) The fiber-1, fiber-1-Δ1 and fiber-1-Δ2 proteins were localized mainly in the cytoplasm of LMH cells (60×), the scale bar was 20 µm. (**E**) Prediction of cellular sublocalization of fiber-1, fiber-1-Δ1 and fiber-1-Δ2 proteins.

**Figure 3 microorganisms-12-02265-f003:**
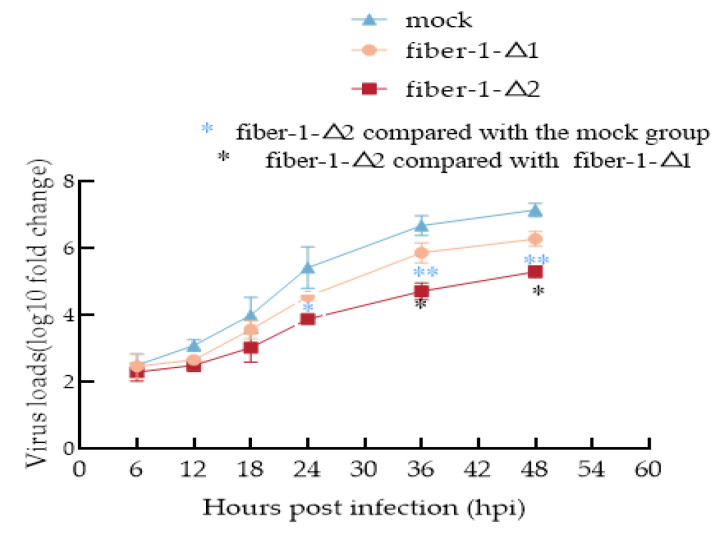
Fiber-1-Δ1 and fiber-1-Δ2 inhibit FAdV-4 replication in LMH cells. LMH cells were transfected with pEF1α-HA, pEF1α-HA-fiber-1-Δ1 or pEF1α-HA-fiber-1-Δ2 and then infected with FAdV-4 (MOI = 0.01). Viral replication was determined at 6, 12, 18, 24, 36, and 48 hpi via absolute quantitative PCR. The results are expressed as the means ± SDs of three independent experiments. pEF1α-HA-fiber-1-Δ2 compared with the mock group, significant differences are denoted by blue* pEF1α-HA-fiber-1-Δ2 compared with pEF1α-HA-fiber-1-Δ1, significant differences are denoted by black*. One * indicates *p* < 0.05, and two * indicate *p* < 0.01.

**Figure 4 microorganisms-12-02265-f004:**
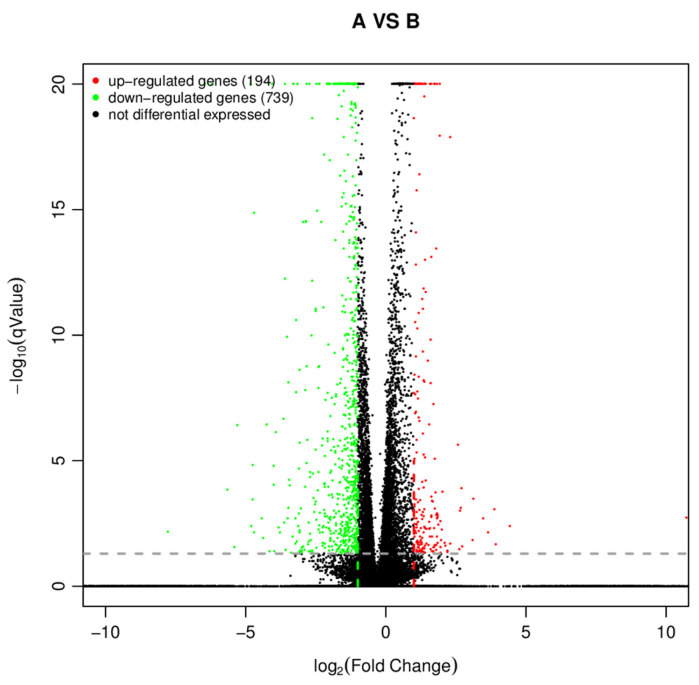
DEGs between fiber-1-Δ1 (A) and fiber-1-Δ2 (B) overexpression in LMH cells. The red dots (●) indicate upregulated genes and the green dots (●) indicate downregulated genes. The black dots (●) indicate that the genes are not differentially expressed.

**Figure 5 microorganisms-12-02265-f005:**
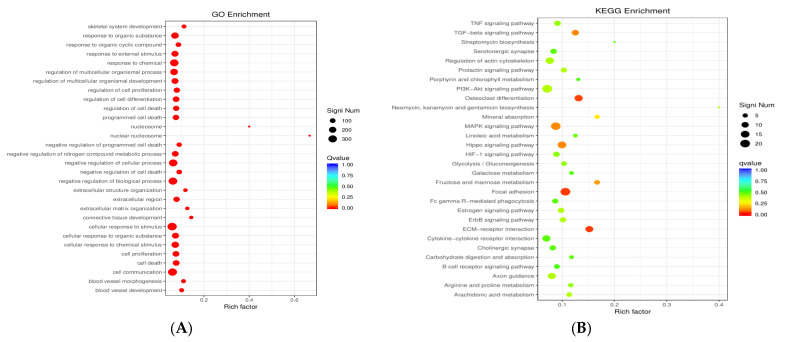
GO and KEGG enrichment analyses of DEGs. (**A**) The top 30 enriched GO categories. (**B**) The top 30 enriched KEGG categories. The vertical axis represents the functional annotation information, and the horizontal axis represents the enrichment factor corresponding to the function (the number of DEGs annotated to the function divided by all the genes). In terms of the number of genes annotated to a function, the size of the q value is indicated by the color of the dot, so the smaller the q value, the redder it is. The number of DEGs associated with each function is indicated by the size of the dots (only the 30 GO terms with the highest enrichment were selected).

**Figure 6 microorganisms-12-02265-f006:**
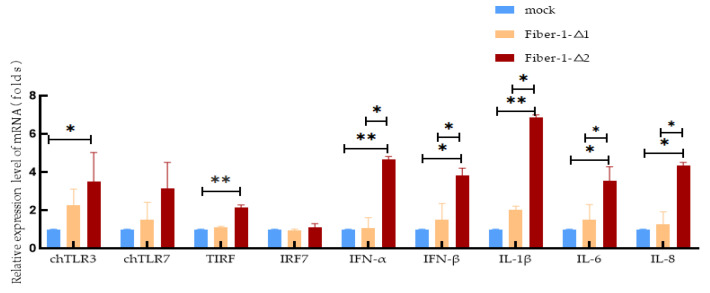
Effects of fiber-1-Δ1 and fiber-1-Δ2 on the expression of innate immune signaling pathway-related genes during FAdV-4 infection. The results are expressed as the means ± SDs of three independent experiments. * indicates *p* < 0.05 and ** indicates *p* < 0.01. In the mock group, LMH cells were transfected with the pEF1α-HA plasmid; in the fiber-1-Δ1 group, LMH cells were transfected with the pEF1α-HA-fiber-1-Δ1 plasmid; and in the fiber-1-Δ2 group, LMH cells were transfected with the pEF1α-HA-fiber-1-Δ2 plasmid.

**Table 1 microorganisms-12-02265-t001:** Fiber-1-Δ1 and fiber-1-Δ2 gene cloning primers.

Primer Name	Sequence (5′-3′)	Amplified Sequence Length
Fiber-1-Δ1-U	CCG*GAATTC*GGATGGGTGGCGGAGGAGGAGGT	399 bp
Fiber-1-Δ1-D	CCC*GGTACC*TCAGGGTCCCACGGAGCTG	
Fiber-1-Δ2-U	CCG*GAATTC*GGTCAGGGTCCCACGGAGCTG	606 bp
Fiber-1-Δ2-D	CCC*GGTACC*TCAGATTGGGCCCGTGGTCA	

Fiber-1-Δ1-F (217–615 bp). Fiber-1-Δ2-R (631–1236 bp).

**Table 2 microorganisms-12-02265-t002:** Primers used for the qPCR.

Primer Name	Sequence (5′-3′)	Login Number
TLR3-U	ACAATGGCAGATTGTAGTCACCT	NM_001011691.3
TLR3-D	GCACAATCCTGGTTTCAGTTTAG	
TLR7-U	TCTGGACTTCTCTAACAACA	NM_001011688
TLR7-D	AATCTCATTCTCATTCATCATCA	
IFN-α-U	ATGCCACCTTCTCTCACGAC	AB021154.1
IFN-α-D	AGGCGCTGTAATCGTTGTCT	
INF-β-U	CCTCAACCAGATCCAGCATT	KF741874.1
INF-β-D	GGATGAGGCTGTGAGAGGAG	
IL-1β-U	GTTAATGATGAAGATGTTGATAGC	NM_204305.1
IL-1β-D	GTTCCAGACACAGCAATC	
IL-6-U	TGGTGATAAATCCCGATGAAG	NM_204628.2
IL-6-D	GGCACTGAAACTCCTGGTCT	
IL-8-U	CCATCTTCCACCTTTCACA	HM179639.1
IL-8-D	ATCCCACAGCACTGACCAT	
TRIF-U	AGCCTGATGGAGAGAGACAGAG	NM_001081506.1
TRIF-D	GATAGACGAGAGGAACTGACCTG	
IRF7-U	ACACTCCCACAGACAGTACTGA	NM_205372.1
IRF7-D	TGTGTGTGCCCACAGGGTTG	
Hexon-U	ACGATCAGACCTTCGTGGAC	KY379035.1
Hexon-D	GGTGTGCGAGAGGTAGAAGC	
GADPH-U	GCACTGTCAAGGCTGAGAACG	KC294567
GADPH-D	GATGATAACACGCTTAGCACCAC	

**Table 3 microorganisms-12-02265-t003:** Prediction of the B-cell epitopes.

Antigen	Position	Antigenic Peptide	Score
fiber-1-Δ1	85–104	LDSVTGVLKVLVDSQGPLQA	1.208
	116–130	QDFVVNNGVLALASSPSSCLQD	1.148
	32–46	PIYVSDRAVSLLIDD	1.141
	57–66	ALMVKTAAPL	1.095
	9–15	QIAVDPD	1.080
	17–27	PLELTGDLLTLE	1.044
fiber-1-Δ2	82–95	EVNLSLIVPPTVSP	1.172
	4–12	YEVTPVLGI	1.145
	29–55	IGYYIYMVSSAGLVNGLITLELAHDLT	1.140
	171–182	DAIAFTVSLPQT	1.123
	110–119	DVGYLGLPPH	1.122
	98–106	QNHVFVPNSF	1.117
	68–78	NFTFVLSPMYP	1.144
	153–159	LGYCAAT	1.111
	124–139	WYVPIDSPGLRLVSFM	1.105
	193–199	PDTVVTT	1.1096

**Table 4 microorganisms-12-02265-t004:** Canonical pathways related to the immune response in the fiber-1-Δ1 and fiber-1-Δ2 groups.

Pathway ID	KRGG Pathway	Downregulated Genes	Upregulated Genes
ko04151	PI3K–Akt signaling pathway	ITGA8, ITGA7, EPHA2, COL9A2, ITGB3, ITGB4, ITGB6, TNR, GNG2, CSF3R, PIK3R3, PDGFB, OSMR, FN1, CDKN1A, GNG10, TNC, LAMC2, PIK3AP1, THBS1, PPP2R2B	GYS2, PPP2R2C
ko04010	MAPK signaling pathway	NFKB2, FOS, MRAS, JUND, FLNC, MAP3K12, JUN, PDGFB, PTPN5, FLNB, LOC107055388, HSPA2, TGFB1, TGFB2, DUSP10, PRKCB, HSPA8, DUSP8, DUSP4	CD36, GYS2, PPP2R2C
ko04060	Cytokine–cytokine receptor interaction	BMP7, TNFRSF21, CSF3R, PDGFB, OSMR, IL8L2, TNFRSF8, TGFB1, TGFB2, TNFSF15, CCL4, CXCL14, LOC10705392	
ko04024	cAMP signaling pathway	FOS, JUN, PIK3R3, PTGER2, SOX9, HCN2, ATP2B4	NPY, GABBRR2, CAMK4
ko04350	TGF-beta signaling pathway	NBL1, BMP7, SMAD6, TGFB1, TGFB2, CDKN2B, ID4, ID1, GDF6, THBS	
ko04630	Jak–STAT signaling pathway	CISH, CSF3R, PIK3R3, OSMR, CDKN1A, SOCS3, SOCS2, PIM1, STAT3	
ko04668	TNF signaling pathway	FOS, CEBPB, EDN1, JUN, PIK3R3, MLKL, PTGS2, SOCS3	
ko04620	Toll-like receptor signaling pathway	FOS, JUN, PIK3R3, IL8L2, CCL4	
ko04310	Wnt signaling pathway	CTBP2, JUN, WNT5A, APC2, PRKCB	
ko04621	NOD-like receptor signaling pathway	RIPK3, IL8L2	
ko046222	RIG-I-like receptor signaling pathway	IL8L2	

## Data Availability

The original contributions presented in the study are included in the article; further inquiries can be directed to the corresponding author.

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
