# Peer review of "The Knob Domain of the Fiber-1 Protein Affects the Replication of Fowl Adenovirus Serotype 4"

_microorganisms, 2024, doi:10.3390/microorganisms12112265_

Round 1
Reviewer 1 Report
Comments and Suggestions for Authors
The authors investigate and present the role of knob domain of the fiber-1 protein of fowl adenovirus-4. Investigating the role of different parts of the fiber-1 protein by expressing truncated parts of the fiber-1 showed that it localized in the cytoplasm of LMH cells in agreement of software predictions. Further, the knob domain was demonstrated to inhibit actual viral replication. Authors also analyzed immune pathways affected by the truncated proteins.
Specific comment:
1)All experiments need a critical control of full length of fiber-1 protein or actual virus infection in protein localization study.
2)The concept of inhibition of viral replication by truncated proteins is immature and cannot be compared to immunity.
Author Response
- All experiments need a critical control of full length of fiber-1 protein or actual virus infection in protein localization study.
Response:Thank you for pointing this out. We agree with this comment. we carefully considered your comment and provide details below:
To verify the localisation of Fiber-1-Δ1 and Fiber-1-Δ2 proteins, we simultaneously transfected full-length Fiber-1 protein into LMH cells, and the results showed that Fiber-1 protein was expressed in the cytoplasm (Figure 2D), we analyzed the amino acid sequences of Fiber-1 online (https://www.genscript.com/psort.html, accessed on 24 August 2024), Fiber-1 protein was expressed in the cytoplasm with a reliability of 70.6% (Figure 2E), which is consistent with our immunofluorescence results, which can be seen in Line258-266 of the manuscript.
Meanwhile, we performed immunofluorescence assay to detect protein sublocalisation after recombinant plasmid transfection and viral infection, which can be seen in Line120-122 of the manuscript.
Figure 2. (D). The Fiber-1, Fiber-1-â–³1 and Fiber-1-â–³2 proteins were localized mainly in the cytoplasm of LMH cells (60×). (E). Prediction of cellular sublocalisation of Fiber-1, Fiber-1-Δ1 and Fiber-1-Δ2 proteins.
- The concept of inhibition of viral replication by truncated proteins is immature and cannot be compared to immunity
Response:Thank you for pointing this out. We agree with this comment. Therefore,
we answer as follows:
As the Fiber-1 protein of FAdV-4 consists of three domains, the knob, shaft and tail, A Masters student in our lab transfected full-length Fibre-1 protein into LMH cells and found that the expression levels of TLR 1b, TLR 3, TLR 7 and TLR 21 mRNAs were significantly upregulated, as well as the expression levels of the pathway-related molecules IRF3/7 and IFN-β.
In this manuscript, we hope to explore which part of the Fiber-1 protein plays a key role, which will help us to further understand the role of Fiber-1 protein and provide a theoretical basis for adenovirus subunit vaccines. Fiber-1-Δ2 is much smaller than full-length Fiber-1,which facilitates the in vitro expression of FAdV-4 used to study the FAdV-4-host relationship.
Truncated proteins have been used in a variety of virus-related studies to explore their functions and develop diagnostic reagents and vaccines. In Mingliang Zhao's experiments, truncated proteins were also used to study the induction of LMH apoptosis by the PX protein of FAdV4, and the precise alanines 11 and 129 of PX are critical for PX-induced apoptosis(Zhao M, Duan X, Wang Y, Gao L, Cao H, Li X, Zheng SJ. A Novel Role for PX, a Structural Protein of Fowl Adenovirus Serotype 4 (FAdV4), as an Apoptosis-Inducer in Leghorn Male Hepatocellular Cell. Viruses. 2020 Feb 18;12(2):228. doi: 10.3390/v12020228. PMID: 32085479; PMCID: PMC7077197).
Although there are fewer studies on virus inhibition by truncated proteins, and most of these studies are at the laboratory exploratory stage, we believe that with the development of biotechnology, the structure and function of the proteins will become clearer, and the natural immunity induced by viral structural proteins will be better explained.

Reviewer 2 Report
Comments and Suggestions for Authors
This manuscript presents an interesting and well-executed study investigating the roles of different domains of the Fiber-1 protein in FAdV-4 infection. The research methodology, which includes using LMH cells, transcriptome sequencing, and detailed functional analysis of DEGs, is appropriate and robust. Therefore, the results, which support the authors' hypothesis, are highly reliable. This approach provides valuable insights into the interaction between FAdV-4 and its host, instilling confidence in the validity of the findings.
Minor observations:
Abbreviations: The manuscript would benefit from ensuring that the full name is mentioned before introducing abbreviations, especially for readers needing to become more familiar with certain terms. This will enhance clarity and accessibility.
Figure 3: Using asterisks in the line chart, particularly when representing statistical significance, can be confusing. It is presumed that the double asterisks (**) refer to Fiber-1-â–³2; however, this should be explicitly defined in the figure legend to avoid ambiguity and ensure clear communication of the results.
Protein Structure Visualization: Introducing a figure that illustrates the complete Fiber-1 protein, highlighting the peptide fragments (Fiber-1-â–³1 and Fiber-1-â–³2), and mapping the domains and identified epitopes would significantly improve the reader's understanding of the research. This could also aid in visualizing the structural relationships between the domains and their functional impacts.
Epitope Prediction Method: The manuscript could benefit from further clarification on the rationale behind using in silico methods for epitope prediction. While not as definitive as in vitro or in vivo testing, these methods are useful for initial identification. However, in vitro or in vivo immunogenicity testing could provide more reliable data. Purifying the Fiber-1 protein fragments followed by immunogenicity testing would likely yield more robust conclusions regarding their potential for inducing an immune response. This discussion will help the reader understand the thought process behind the research methods and the potential limitations of the findings.
Discussion: The manuscript could benefit from a more comprehensive discussion of the broader implications of their findings. Specifically, a paragraph could be added addressing how the identification of the Fiber-1-â–³2 domain's role in viral replication and immune modulation could impact future developments in vaccine design, diagnostic tools, and understanding of virus-host interactions. This discussion will provide important perspectives on the significance of these findings and help the reader understand the potential practical applications of the research, inspiring hope for the future of virology and molecular biology.
Author Response
1 Abbreviations: The manuscript would benefit from ensuring that the full name is mentioned before introducing abbreviations, especially for readers needing to become more familiar with certain terms. This will enhance clarity and accessibility.
Response: Thank you for pointing this out. We agree with this comment. we answer as follows:
PI3K‒Akt(phosphoinositide 3-kinase-protein kinase B,PI3K‒Akt),Line26
MAPK(mitogen-activated protein kinase, MAPK),Line27
Jak-STAT {JAK(janus tyrosine kinase)-STAT(signal transducer and activator of transcription),Jak-STAT } , Line28-29
NOD-like receptors (nucleotide binding oligomerization domain(NOD)- like receptors,NLRs)), Line29-30
MDA5(melanoma differentiation-associated gene 5,MDA5, Line76
cGAS(cyclic guanosine monophosphate-adenosine monophosphate synthase, cGAS), Line77
MyD88(myeloid differentiation factor 88, MyD88), Line89
SDS( sodium dodecyl sulfate,SDS),Line128-129
SDS‒PAGE(sodium dodecyl sulfate polyacrylamide gel electropheresis SDS‒PAGE), Line130-131
PBST(phosphate-buffered saline with tween-20,PBST, Line137
BSA(bovine serum albumin,BSA),Line148
FITC(fluoresceine isothiocyanate,FITC), Line152
DAPI(4',6-diamidino-2-phenylindole,DAPI), Line154,
TNF(tumor necrosis factor), Line330-331,
TGF-β (transforming growth factor-beta, TGF-β), Line331
RIG-I-like receptor(RIG-â… -like helicase receptors,RLRs, Line 345-346
RIPK3(receptor interacting protein kinase 3,RIPK3 ), Line351
IBH (inclusion body hepatitis,IBH),Line403
2 Figure 3: Using asterisks in the line chart, particularly when representing statistical significance, can be confusing. It is presumed that the double asterisks (**) refer to Fiber-1-â–³2; however, this should be explicitly defined in the figure legend to avoid ambiguity and ensure clear communication of the results.
Response: Thank you for pointing this out. We agree with this comment. we answer as follows:
We have illustrated in the legend in Figure 3 that: Fiber-1-△2 compared with the mock group denoted by *, Fiber-1-△2 compared with Fiber-1-△1 denoted by *. Details can be found in Figure 3
Figure 3. Fiber-1-â–³1 and Fiber-1-â–³2 inhibit FAdV-4 replication in LMH cells
3 Protein Structure Visualization: Introducing a figure that illustrates the complete Fiber-1 protein, highlighting the peptide fragments (Fiber-1-â–³1 and Fiber-1-â–³2), and mapping the domains and identified epitopes would significantly improve the reader's understanding of the research. This could also aid in visualizing the structural relationships between the domains and their functional impacts.
Response: Thank you for pointing this out. We agree with this comment.
We have included an image of the complete Fiber-1 protein with the positions of Fiber-1-Δ1 and Fiber-1-Δ2 labelled in the figure. Meanwhile the red boxes represent the highest scoring antigenic epitopes .Details are shown in Figure 1D.
Figure 1. (D). Complete Fibre-1 protein map and prediction of the hydrophilicity, antigenicity, flexibility and surface accessibility of potential dominant B-cell epitopes of Fiber-1-â–³1 and Fiber-1-â–³2.Red boxes represent the highest scoring antigenic epitopes.
4 Epitope Prediction Method: The manuscript could benefit from further clarification on the rationale behind using in silico methods for epitope prediction. While not as definitive as in vitro or in vivo testing, these methods are useful for initial identification. However, in vitro or in vivo immunogenicity testing could provide more reliable data. Purifying the Fiber-1 protein fragments followed by immunogenicity testing would likely yield more robust conclusions regarding their potential for inducing an immune response. This discussion will help the reader understand the thought process behind the research methods and the potential limitations of the findings.
Response:Thank you for pointing this out. We agree with this comment.Therefore,
after careful consideration,the discussion is as follows
“In our study, to determine the antigenic peptides of Fiber-1-Δ1 and Fiber-1-Δ2, we used in silico methods for epitope prediction. Antigenic epitope, also known as antigenic site, is a special chemical group on the surface of antigen molecules that determines the specificity of the antigen and is the region where the antigen interacts with the antibody to elicit an immune response. We have predicted the hydrophilicity, antigenicity, flexibility and surface accessibility of the potentially dominant antigenic epitopes in B cells for both Fiber-1-Δ1 and Fiber-1-Δ2 and may provide key information for the establishment of novel diagnostic methods and the development of related vaccines. This method is not as definitive as in vitro or in vivo testing, which is a shortcoming of this manuscript.”The details can be found in Line 430-439
5 Discussion: The manuscript could benefit from a more comprehensive discussion of the broader implications of their findings. Specifically, a paragraph could be added addressing how the identification of the Fiber-1-â–³2 domain's role in viral replication and immune modulation could impact future developments in vaccine design, diagnostic tools, and understanding of virus-host interactions. This discussion will provide important perspectives on the significance of these findings and help the reader understand the potential practical applications of the research, inspiring hope for the future of virology and molecular biology.
Response:Thank you for pointing this out. We agree with this comment.Therefore,
after careful consideration,the discussion is as follows:
“From the above results, it can be seen that Fiber-1-Δ2 activated the natural immune signalling pathway to inhibit the replication of FAdV-4 in LMH cells. In this experiment, Fiber-1-Δ2 was obtained by truncating Fiber-1, which is rich in B cell antigenic epitopes, the key to successful vaccine development. At the same time, the truncated protein can effectively avoid unnecessary antigenic components and reduce potential toxic reactions. Meanwhile, Fiber-1-Δ2 may have better antigenicity than full-length Fiber-1 and can be used as a detection antigen for the development of the FAdV-4 ELISA antibody assay with higher sensitivity to react with specific antibodies, development of more valuable assays and more to understand all aspects of viral pathogenesis in order to propose novel antiviral strategies.Finally, Fiber-1-Δ2 is much smaller than full-length Fiber-1, which facilitates the in vitro expression of FAdV-4 used to study the FAdV-4-host relationship.”The details can be found in Line 535-546.

Round 2
Reviewer 2 Report
Comments and Suggestions for Authors
All suggestions were accepted, and the changes introduced in the text were suggested so the manuscript could be accepted for publication. However, we recommend that the authors modify the new paragraph (Lines 419-428) introduced below for better suitability.
"In our study, we utilized in silico methods to predict the epitopes/peptides of Fiber-1-Δ1 and Fiber-1-Δ2. These antigenic sites, or epitopes, are specific chemical groups on the surface of antigen molecules that activate the immune system and facilitate antibody interactions. We evaluated key characteristics of the potentially dominant B cell epitopes for both Fiber-1-Δ1 and Fiber-1-Δ2, including hydrophilicity, antigenicity, flexibility, and surface accessibility. Our findings may provide valuable insights for the development of novel diagnostic methods and related vaccines. However, it is important to note that this in silico approach is less definitive than in vitro or in vivo testing, which represents a limitation of this study."
Author Response
All suggestions were accepted, and the changes introduced in the text were suggested so the manuscript could be accepted for publication. However, we recommend that the authors modify the new paragraph (Lines 419-428) introduced below for better suitability.
"In our study, we utilized in silico methods to predict the epitopes/peptides of Fiber-1-Δ1 and Fiber-1-Δ2. These antigenic sites, or epitopes, are specific chemical groups on the surface of antigen molecules that activate the immune system and facilitate antibody interactions. We evaluated key characteristics of the potentially dominant B cell epitopes for both Fiber-1-Δ1 and Fiber-1-Δ2, including hydrophilicity, antigenicity, flexibility, and surface accessibility. Our findings may provide valuable insights for the development of novel diagnostic methods and related vaccines. However, it is important to note that this in silico approach is less definitive than in vitro or in vivo testing, which represents a limitation of this study."
Response:Thank you for pointing this out. We agree with this comment.Therefore,
after careful consideration,we have modified it as follows:
“In our study, we utilized in silico methods to predict the epitopes/peptides of Fiber-1-Δ1 and Fiber-1-Δ2. These antigenic sites, or epitopes, are specific chemical groups on the surface of antigen molecules that activate the immune system and facilitate antibody interactions. We evaluated key characteristics of the potentially dominant B cell epitopes for both Fiber-1-Δ1 and Fiber-1-Δ2, including hydrophilicity, antigenicity, flexibility, and surface accessibility. Our findings may provide ideas to explain the inhibition of viral replication by the Fiber-1-Δ2 in terms of natural immunity, as well as valuable insights for the development of novel diagnostic methods and related vaccines. However, it is important to note that this in silico approach is less definitive than in vitro or in vivo testing, which represents a limitation of this study."The details can be found in Line 430-440.
